# Insight into the Phytoremediation Capability of *Brassica juncea* (v. Malopolska): Metal Accumulation and Antioxidant Enzyme Activity

**DOI:** 10.3390/ijms20184355

**Published:** 2019-09-05

**Authors:** Arleta Małecka, Agnieszka Konkolewska, Anetta Hanć, Danuta Barałkiewicz, Liliana Ciszewska, Ewelina Ratajczak, Aleksandra Maria Staszak, Hanna Kmita, Wiesława Jarmuszkiewicz

**Affiliations:** 1Department of Biotechnology, Institute of Molecular and Biotechnology, Adam Mickiewicz University, Collegium Biologicum, Uniwersytetu Poznańskiego 6, 61-614 Poznań, Poland; 2Department of Biochemistry, Institute of Molecular Biology and Biotechnology, Adam Mickiewicz University, Collegium Biologicum, Uniwersytetu Poznańskiego 6, 61-614 Poznań, Poland; 3Department of Trace Element Analysis by Spectroscopy Method, Faculty of Chemistry, Adam Mickiewicz University, Uniwersytetu Poznańskiego 8, 61-614 Poznań, Poland; 4Institute of Dendrology, Polish Academy of Sciences, Parkowa 35, 62-035 Kórnik, Poland; 5Plant Physiology Department, Institute of Biology, University of Bialystok, Ciołkowskiego 1J, 15-245 Bialystok, Poland; 6Department of Bioenergetics, Institute of Molecular and Biotechnology, Adam Mickiewicz University, Collegium Biologicum, Uniwersytetu Poznańskiego 6, 61-614 Poznań, Poland

**Keywords:** oxidative stress, antioxidative system, Brassicaceae family, heavy metals

## Abstract

Metal hyperaccumulating plants should have extremely efficient defense mechanisms, enabling growth and development in a polluted environment. *Brassica* species are known to display hyperaccumulation capability. *Brassica juncea* (Indiana mustard) v. Malopolska plants were exposed to trace elements, i.e., cadmium (Cd), copper (Cu), lead (Pb), and zinc (Zn), at a concentration of 50 μM and were then harvested after 96 h for analysis. We observed a high index of tolerance (IT), higher than 90%, for all *B. juncea* plants treated with the four metals, and we showed that Cd, Cu, Pb, and Zn accumulation was higher in the above-ground parts than in the roots. We estimated the metal effects on the generation of reactive oxygen species (ROS) and the levels of protein oxidation, as well as on the activity and gene expression of antioxidant enzymes, including superoxide dismutase (SOD), catalase (CAT), and ascorbate peroxidase (APX). The obtained results indicate that organo-specific ROS generation was higher in plants exposed to essential metal elements (i.e., Cu and Zn), compared with non-essential ones (i.e., Cd and Pb), in conjunction with SOD, CAT, and APX activity and expression at the level of encoding mRNAs and existing proteins. In addition to the potential usefulness of *B. juncea* in the phytoremediation process, the data provide important information concerning plant response to the presence of trace metals.

## 1. Introduction

Trace metal element contamination in soils is one of the world’s major environmental problems, posing significant risks to human health, as well as to ecosystems ([1]). Metals such as zinc (Zn), iron (Fe), and copper (Cu) are essential micronutrients required for a wide range of physiological processes in all plant organs, and the processes are based on the activities of various metal-dependent enzymes and proteins. However, they can also be toxic at elevated levels. Metals such as arsenic (As), mercury (Hg), cadmium (Cd), and lead (Pd) are nonessential and potentially highly toxic [2]. Trace metal element toxicity includes changes in the chlorophyll concentration in leaves and damage of the photosynthetic apparatus, inhibition of transpiration, and destruction of carbohydrate metabolism, as well as nutrition and oxidative stress, which collectively affect plant development and growth [3,4,5,6,7].

Biological organisms are incapable of degrading metals, so they persist in their body parts and environment, leading to health hazards [8]. Metal accumulation and other abiotic stresses cause excess reactive oxygen species (ROS) generation, leading to oxidative stress [7]. Plant cells are equipped with enzymatic mechanisms to eliminate or reduce oxidative damage that occurs under metal accumulation. The antioxidative defense system includes superoxide dismutase (SOD), catalase (CAT), and ascorbate peroxidase (APX), which are regarded as responsible for maintaining the balance between ROS production and scavenging [9].

The Brassicaceae family includes many genera abundant in metallophytes, such as *Thlaspi*, *Brassica,* and *Arabidopsis*. They accumulate a wide range of heavy metals, especially Zn, Cd, nickel (Ni), thallium (Tl), chromium (Cr), and selenium (Se) [10]. The term hyperaccumulator is used for plants that accumulate 1000 mg per kg of dry matter of any above-ground tissue when grown in their natural habitat [11,12]. As of 2013, approximately 500 metal hyperaccumulator plant species were described [13,14], and the number is increasing. *B. juncea* exhibits some traits of a metal hyperaccumulator—this species can take up significant quantities of Pb, Cd [15,16], Cr, Cu, Ni, Pb, and Zn [10,17], although its translocation ability is not as efficient as shown for other known hyperaccumulators. Metal hyperaccumulating plants should have extremely efficient defense mechanisms, enabling growth and development in a polluted environment. Therefore, the objective of the present study was to estimate the contribution of the *B. juncea* (v. Malopolska) enzymatic antioxidant system to combating the oxidative stress induced by essential (Cu, Zn) and non-essential (Pb, Cd) metal elements to allow survival under adverse environmental conditions. The analysis included trace metal accumulation, level of stress parameters, and antioxidant enzyme activity, as well as estimation of encoding mRNA and enzyme protein levels.

## 2. Results

### 2.1. Levels of Metal Accumulation

Research using laser ablation combined with plasma mass spectrometry (LA-ICP-MS) made it possible to determine the levels of metal accumulation in *B. juncea* organs (Figure 1). The analyses were performed for roots, stems, and leaves. In the case of roots, Pb constituted approximately 60% of all accumulated metals. In addition, approximately 4 times higher levels of accumulated Cu and Zn, as well as more than 140 times higher levels of Cd, were found in roots compared to control plant seedlings. In the stems and leaves, high levels of Cu and Zn were observed to be approximately 20 times higher than in control plants. The data allowed for calculation of the amount of accumulated Cu, Cd, Zn, and Pb in the above-ground parts, which were 58%, 55%, 52%, and 38% higher, respectively, than the amount in the roots. The results indicate that *B. juncea* is a good accumulator of trace metals, especially Cd.

### 2.2. Biomass and Morphological Changes

The metals used in the research did not dramatically increase *B. juncea* (v. Malopolska) seedling biomass (Figure 2). The highest inhibition of biomass growth was observed for seedlings exposed to Cu. After 96 h of treatment, the seedling biomass was approximately 34% lower than that of control plants. The weakest effect was observed for seedlings treated with Pb, as after 96 h of treatment, the seedlings were approximately 10% lighter compared to control plants. The metals used in the study also did not appreciably inhibit the increase in root length. The value of the index of tolerance (IT), based on average root length also did not change dramatically (Figure 2). After 96 h of treatment, we observed the lowest IT value for Pb (70%) and the highest IT value for Cd, i.e., 90,4%. We observed the occurrence of necrotic spots on leaves and the inhibition of leaf blade surface growth with respect to control seedlings in the above-ground parts of seedlings. Moreover, in Cd-treated seedlings, leaves were slightly twisted, whereas Cu caused strong chlorosis and shortening of the end of leaves. The smallest morphological changes were observed for seedlings treated with Zn.

### 2.3. Production and Localization of ROS

The metal-treated seedlings increased O_2_^.-^ production at levels comparable for shoots and roots compared to control seedlings, but the fluctuation in the production observed for control plants was maintained (Figure 3). In the roots, the highest values were mainly observed in the first 72 h (over 30%), whereas in the above-ground parts, the highest values were observed for 48 h (over 30–40%). After 96 h, the levels of O_2_^.−^ decreased, which may indicate high activity of the SOD enzyme. The highest level of O_2_^.−^ in roots was observed for plants treated with Zn compared with shoots treated with Zn and Cd.

The profile of the changes in the H_2_O_2_ level was similar for control roots and shoots, but the levels were distinctly higher in roots. The highest H_2_O_2_ amount was observed in roots treated with Cu, Cd, and Zn. For metal-treated samples, a significant increase in H_2_O_2_ occurred between 48 and 72 h of treatment, and the observed profile of H_2_O_2_ changes was more homogenous for shoots. We noticed a large difference in the level of H_2_O_2_ in roots after 96 h of treatment, reaching approximately 20–50% higher compared to the control. As in the case of O_2_^.−^, H_2_O_2_ levels were also confirmed by confocal microscopy (Figure 4). The most intensive fluorescence DHE, indicating the presence of O_2_^.−^, was observed for the *B. juncea* roots treated for 24 h with 50 μM Cd and Zn. The highest amount of H_2_O_2_ generated was observed in roots treated with 50 μM Cu, Cd, and Zn (Figure 5).

### 2.4. Levels of Oxidized Proteins

The levels of protein oxidative modification imposed by the metal treatment were 12–44% higher for roots and above-ground parts compared to control plants (Figure 1). The level of oxidized proteins reached a maximum after 48 h and was 3-fold higher than in the shoots of control plants.

### 2.5. Enzyme Antioxidant Activity

SOD activities were 25–50% higher in the roots of plants treated with trace metals. In the above-ground parts, greater differences in SOD activity between research variants, ranging from 8 to 70%, were observed. However, the general activity of SOD was higher in roots and shoots compared to control seedlings (Figure 3) and changed differently for the seedling parts. In the case of roots, the activity level and profile were comparable to those of control seedlings, whereas for shoots, after the initial increase, the activity decreased significantly after 96 h. The generation of H_2_O_2_ caused a rapid increase in CAT activity within 24 h of cultivation, i.e., from 30% to 70% in the roots of plants treated with trace metals, especially in plants treated with Zn (Figure 4). In the next days, we observed a slight decrease (approximately 12–55%), but this decrease remained higher than that in control plants. The highest CAT activity was observed above ground in the first 48 h of cultivation (56%) in plants exposed to Cd. Activities of APX, a second enzyme involved in the dismutation of hydrogen peroxide, systematically increased in roots exposed to metals during the cultivation period, especially in plants grown in the presence of Cu and Zn, which had approximately 10–43% higher levels than those observed in the control (Figure 4). In the above-ground parts of *B. juncea* cultured in the presence of trace metals, we observed an increase in the intensity of APX during the first 48 h, reaching a maximum in plants treated with Cd for 48 h, approximately 62% higher than in the control, and then a slight decrease, but the activities were approximately 2-fold higher than those in control plants. The activity profiles of CAT and APX differed between the control roots and shoots (Figure 4). The metal treatment increased the activity of both enzymes, and the CAT activity profile appeared to be maintained in roots and shoots. However, the APX profile did not differ from that of the control plants with respect to treated shoots, whereas in treated roots, the APX activity profile was variable and metal-dependent, although comparable for Cu and Zn.

### 2.6. Levels of Gene Transcripts

To estimate possible changes at the level of CuZnSOD and MnSOD encoding gene transcripts, we used an electrophoretic separation technique and the CpAtlas program (Figure 6). In the case of CuZnSOD, a decrease in the expression of the gene encoding CuZnSOD was observed in the roots of plants treated with trace metals after 4 h and 24 h of cultivation, with the exception of the roots of *B. juncea*-treated Cu. Induction of the gene in the above-ground parts was visible, with an approximate 2-fold increase in the level of the transcript in plants after 8 h of copper treatment and an approximate 2-fold decrease in plants after 4 h of zinc treatment. The results indicate that the presence of cadmium ions had no significant effect on the induction of CuZnSOD gene expression because no significant changes in the level of the transcripts was observed in either the roots or above-ground parts of *B. juncea* plants.

When analyzing changes in the expression of the gene encoding MnSOD, a decrease in the expression was observed in the roots and above-ground parts of plants after 4 h of treatment with lead ions; in the remaining research variants, there were no significant differences in transcript levels compared to control plants. An approximate 2-fold increase in the level of the transcript was found in plant roots after 24 h of Pb and Zn treatment in comparison to the control. The greatest decrease in expression was observed after 24 h in the above-ground parts of plants treated with Cu, which was almost 5-fold higher than that in the control (Figure 6).

### 2.7. Identification of Enzyme Forms

To distinguish between the enzyme forms, Western blot analysis was performed for protein extracts from roots and above-ground seedling parts in the absence and presence of the metal treatment (Figure 7). This allowed for the detection of MnSOD (25 kDa) and CuZnSOD (15 kDa and 20 kDa) subunits. The obtained signal was similar for both the treated and control seedlings. Thus, the metal presence likely did not change the levels of the CuZnSOD and MnSOD proteins.

## 3. Discussion

Trace metals are one of the most important abiotic stress factors affecting the natural environment. As a result of anthropogenic activities, we can observe their increasing levels from year-to-year. Metal toxicity results in effects at physiological and cellular levels, leading to distorted metabolism, including plant metabolism [18]. Abiotic stresses, including the presence of trace metals in soil, are estimated to be the main cause of global crop yield reduction of ca. 70% and thus are considered a great constraint to crop production. This situation has worsened due to disturbed equilibrium between crop production and human population growth. Therefore, it is especially important to understand plant responses to such stress factors. This also applies to trace metals [12]. In the present study, this was clearly visible in the growth of plant biomass, which significantly decreased during the culture in the presence of heavy metals. Copper and zinc ions are essential for the normal growth and development of all organisms but can be toxic to plants at excessive levels. Lead and cadmium are nonessential elements and are toxic to plants even at low levels [8]. Essential and nonessential trace elements, when exceeding the threshold limits, can cause different physiological, morphological, and genetic plant anomalies, including reduced growth, mutations, and increased mortality [8]. Therefore, plants suitable for phytoremediation are, at present, of great importance.

In our study, we noticed that in the case of *B. juncea* v. Malopolska, all the mentioned metals used at 50 μM concentration displayed moderate phytotoxic properties. The biomass increments ranged between 96 mg for Pb-treated plants and 61 mg for Cu-treated plants, and the values were approximately 7% and 41% lower, respectively, than those in control plants. Several studies have shown that high concentrations of trace metals in the soil cause plant growth impairment [9,19]). In *Sesbania drummondii*, a reduction in seedling biomass was caused by Pb—21%, Cu—46.3%, Ni—31.5% and Zn—25.2% [20]. The inhibition of shoot growth by trace metals may be due to a decrease in photosynthesis, as trace metals disturb mineral nutrition and water balance, change hormonal status, and affect membrane structure and permeability [21]. Trace metals might cause an inhibition of root growth that alters water balance and nutrient absorption [12] and decrease calcium uptake in root tips, leading to a decrease in cell division or cell elongation [9,22,23]. According to Marshner [23], Cd-induced mineral stress can reduce plant dry weight accumulation. Other authors have shown a negative influence of Pb [24], Cu [25], Cd [26], and Zn [20]. Despite the inhibitory effect caused by trace metals on the growth of the biomass of *B. juncea,* we observed a high IT amounting to approximately 90% resistance of the plants to trace metals.

The bioaccumulation of trace metals is different for various plant species, reflected by their growth, reproduction, occurrence, and survival in metal-contaminated soil, because the mechanisms of elemental uptake by plants are not the same for all species. The capacity of plants to take up trace metals is different for different metals, and the same trace metal can be accumulated at different ratios in different plant species [27]. Metal bioavailability is also affected by the presence of organic compounds of that metal in plants [8]. The ICP-MS results we obtained indicate that the accumulation of trace metals was higher in above-ground parts than in roots, especially for cadmium, lead, and zinc. The metal concentrations followed an order of Pb > Cu > Zn > Cd in roots, Zn > Cu > Pb > Cd in the stem, and Zn > Cu > Cd > Pb in leaves [28]. Based on the obtained results, it can be concluded that *B. juncea* is a hyperaccumulator of Cd, Zn, and Pb. Cherif and co-authors [29] reported that Zn induced a decrease in Cd uptake and a simultaneous increase in Zn accumulation, indicating a strong competition between these two metals for the same membrane transporters. In our earlier study [28] in *B. juncea* plants treated with a binary combination of metals, namely, PbCu, PbCd, PbZn, CuZn, CuCd, and ZnCd, at a concentration of 25 μM of each, a synergistic response between Zn and Pb was observed, resulting in an increased accumulation of the two metals. The accumulation results obtained for plants treated with Cu are different from those of other researchers. Purakayastha and others [30] showed that Cu is accumulated mainly in above-ground parts of *B. juncea*. This difference may result from different exposure durations of the plant to the metal, other metal concentrations, and different plant ages at the time of analysis of the collected metal. Quaritacci et al. [31] reported that *B. juncea* was identified as a species able to take up and accumulate metals in its above-ground parts, such as Cd, Cu, Ni, Zn, Pb, and Se. It has been observed that this species concentrated Cu, Pb, and Zn in its above-ground parts in amounts much higher than those detected in the metal soluble fractions present in a soil contaminated by acidic water and pyritic slurry [31].

The accumulation of trace metals in organs is dangerous for plants. In an earlier study [32], we confirmed that plants are not adequately protected by the detoxification system because trace metals penetrate in areas with high metabolic activity, such as the cytoplasm, mitochondria, or cell membrane.

The occurrence of oxidation stress conditions in *B. juncea* treated with the trace metals Pb, Cu, Cd, and Zn was confirmed by the increase in the level of oxidized proteins in the roots (approximately 7–12%) and above-ground parts (approximately 13%). Several metals, including Cd, Pb, and Hg, have been shown to cause protein oxidation by depletion of protein thiol groups [33]. ROS cause protein modifications through the formation of carbonyl groups at certain amino acid residues. Such modifications were caused by the presence of heavy metals, e.g., cadmium [34], mercury lead, aluminum, zinc, copper, cobalt, nickel, and chromium [35].

ROS also act as signaling molecules involved in the regulation of many key physiological processes, such as root hair growth, stomatal movement, cell growth, and cell differentiation, when finely tuned and regulated by an antioxidative defense system [12]. We showed an increase in the level of ROS compared to control plants in all plants treated with heavy metals. The O_2_^.−^ rate after 2 h of culture was 2 times higher than that observed in plants grown under control conditions. The high level of O_2_^.−^ was the highest between 24 to 72 h of the treatment depending on the research variant. The highest value of O_2_^−^ was measured in plants treated with Zn, while the highest H_2_O_2_ values were observed in plants treated with Cu and Cd. Similar results were obtained by other researchers. Markovska et al. [36] showed a 10-fold higher level of H_2_O_2_ in the leaves of *B. juncea* after 5 days of treatment with Cd ions at a concentration of 50 μM. Wang et al. [37] observed the highest levels of H_2_O_2_ in *B. juncea* roots treated with Cu ions for 4 days. In our research, the highest level of H_2_O_2_ was obtained after 4 days in plants treated with single metals. The reduction of O_2_^.−^ and the H_2_O_2_ content in roots and above-ground parts of plants treated with trace metals during the cultivation period suggests that some antioxidative enzymes would work effectively in the removal of ROS. To detect ROS in plant cells, we used incubation with fluorescent labels such as 2′7′-difluoroscein and dihydroethidium and imaging under confocal microscopy. We observed increased generation of O_2_^.−^ and H_2_O_2_ in the roots of *B. juncea* treated with trace metals—especially Cd and Zn for O_2_^.−^, and Cu, Cd, and Zn for H_2_O_2_.

The increase in ROS production in metal-treated plants was precisely associated with changes in the activity of antioxidant enzymes. We always observed the induction of antioxidant enzyme activity in *B. juncea* roots and leaves, although there were no significant differences between the used metals. We observed increasing activity of antioxidant enzymes, i.e., 20–158% for SOD, 15–147% for CAT, and 6–68% for APX. The highest activity of SOD in both roots and shoots was observed in plants treated with Zn and Cu. The first line of defense against ROS-mediated toxicity is through SOD, which catalyzes the dismutation of superoxide anions to H_2_O_2_ and O_2_. The stimulation of SOD activity has also been reported in several plants exposed to Pb, Cu, Cd, Zn, Ni, and as ions [20,25,38,39]. We noticed that in the roots of *B. juncea,* the most induced activity of CAT was for Zn, compared with Cd in the above-ground parts. APX was definitely lower than catalase, especially in the above-ground parts, which means that this enzyme complements CAT catalytic activity. APX activity was significantly elevated in the metal-treated plants, which suggests its role in the detoxification of H_2_O_2_. Enhanced CAT and APX activity has been observed in various plant species after application of trace metals: Pb, Cu, Cd, Zn, Ni, and As [20,25,38,39,40]. APX may be responsible for controlling the levels of H_2_O_2_ as signal molecules, and the CAT function is to remove large amounts of oxygen during oxidative stress. APX may be responsible for controlling the levels of H_2_O_2_ as signal molecules, and the CAT function is to remove large amounts of oxygen during oxidative stress [41]. Mohamed et al. [42] showed in *B. juncea* that the higher activity of antioxidant enzymes offers a greater detoxification efficiency, which provides better plant resistance against trace metal-induced oxidative stress. Yadav and co-authors [25] reported increases in the activities of antioxidant enzymes: SOD by 16.2%, DHAR by 27–58%, GR by 35.74%, GST and GPX by 19.19%, and APX by 42.75% in *B. juncea* plants treated with 0.0005 M Cu. The authors indicated that brassinostereoids can regulate the activity of the antioxidant system and help in scavenging overproduced ROS, and can provide tolerance by inducing the expression of regulatory genes such as respiratory burst oxidase homologue, mitogen-activated protein kinase-1, and mitogen-activated protein kinase 3, as well as activating genes involved in antioxidative defense and responses [25]. Other authors [12] have noted that brassinosteroids are a group of hormones that regulate ion uptake in plant cells and reduce trace metal accumulation in plants. An exogenous application of brassinosteroids is widely used to improve crop yield, as well as stress tolerance, in various plant species.

We previously demonstrated an increase in the activity of the antioxidant system at the physiological and biochemical levels. The next step was to determine whether trace metals influence the transcription level of genes encoding suitable defense proteins. ROS concentration at an appropriate level can promote plant development and reinforce resistance to stressors by modulating the expression of a set of genes and redox signaling pathways [12]. In our research, we observed differences in the expression induction depending on the exposure time and the metal used. We observed an increase in the level of the gene coding for CuZnSOD in plants treated with copper, zinc, and lead. The highest level of expression was obtained after 4 h in roots and 8 h in above-ground parts. Romero-Puertas and co-authors [34] noted a drastic reduction in the expression of genes coding for CuZnSOD and no changes in MnSOD in *Pisum sativum* under conditions of stress caused by the presence of Cd. Their results showed a reduction in CuZnSOD levels in the presence of Cd, while in our study, we did not observe significant differences in the level of transcript for plants treated with this metal in relation to control plants. We observed the induction of gene expression encoding MnSOD in *B. juncea* roots after 8 h of exposure to Zn and Pb ions, compared with lead ions in above-ground parts. Other authors did not observe any changes or a low expression of genes coding for SOD, e.g., Fidlago et al. [43] showed no differences in MnSOD-related mRNA accumulation in leaves and roots, but CuZnSOD-related transcripts decreased in leaves but did not change in roots in Cd-treated *Solanum nigrum* L. Others authors [44] indicated that Cd stress induced an upregulated expression of FeSOD, MnSOD, Chl CuZnSOD, Cyt CuZnSOD, APX, GPX, GR, and POD at 4–24 h after treatment began for *Lolium perenne* L., and their results suggested that the gene transcript profile was related to the enzyme activity under Cd stress. Romero-Puertas et al. [34] indicated two groups of genes in pea plants treated with Cd. First, some elements of the signal transduction cascade accentuated or attenuated the Cd effect on CAT, MDHAR, and CuZnSOD mRNA expression. The second was formed by the genes MnSOD, APX, and GR that were not affected by these modulators during the Cd treatment because their expression was not modified compared to control plants.

The effect of Cd on the expression of CuZnSOD was reversed by a nitric oxide (NO)^.^ scavenger, indicating that NO^.^ must be a key element in the regulation of this SOD, showing the existence of a relationship between an increase in ROS production and NO. NO-dependent downregulation was also observed for MnSOD, while the opposite effect was found for APX and GR. This suggests that protein phosphorylation is involved in the response to Cd stress [34]. Bernard and co-authors [45] indicate that molecular analysis (gene expression) is the first level of integration of environmental stressors, and it is supposed to respond to stressors earlier than biochemical markers.

Our results from Western blotting indicate that the presence of trace metals does not increase the synthesis of the proteins CuZnSOD and MnSOD in the organs of *B. juncea* plants, but induces an increase in their activity.

## 4. Materials and Methods

### 4.1. Plant Material

*Brassica juncea* v. Malopolska seeds were grown in Petri dishes for 7 days under optimal conditions. Next, seedlings were cultivated hydroponically on Hoagland’s medium for 7 days in a growth room with a 16/8 h photoperiod, day/night at room temperature and light intensity of 82 μmol m^2^ s^−1^. Then, the applied medium was changed into 100×-diluted Hoagland’s medium and a heavy metal solution in combination; Cu, Pb, Cd, and Zn ions at a concentration of 50 μM were applied. In the cultivation, a solution of Pb(NO_3_)_2_, CuSO_4_, CdCl_2_, Zn SO_4_ was used. The roots and shoots were cut off after 0, 24, 48, 72, and 96 h of cultivation. The roots were dipped sequentially in cold solutions of 10 mM CaCl_2_ and 10 mM EDTA for 5 min each to eliminate trace elements adsorbed at the root surface. Then, roots and shoots were rinsed three times with distilled water, frozen in liquid nitrogen, and stored at −80 °C until molecular analysis.

### 4.2. Phytotoxic Test

The index of tolerance (IT) was calculated according to Wilkins [46]:(1)$$\mathrm{IT}=\frac{\mathrm{average}\;\mathrm{length}\;\mathrm{of}\;\mathrm{roots}\;\mathrm{in}\;\mathrm{tested}\;\mathrm{solution}}{\mathrm{average}\;\mathrm{length}\;\mathrm{of}\;\mathrm{roots}\;\mathrm{in}\;\mathrm{control}}\times 100\%$$

The changes in fresh biomass of control plants and plants treated with metals were measured on a Radwag scale after 0, 24 28, 72, and 96 h of cultivation.

### 4.3. Accumulation of Trace Metals

The determination of trace metal accumulation was performed using inductively coupled plasma mass spectrometry (ICP-MS) model Elan DRC II, (Perkin Elmer Sciex, Concord, Ontario, Canada) connected with laser ablation (LA) model LSX-500 (CETAC Technologies, Omaha, NE, USA). Plant material (roots, stems, and leaves) was rinsed with distilled water, gently dried on blotting paper, weighed, and dried at 70 ± 2 °C. The dried samples were mineralized in an MDS-2000 microwave digestor oven (CEM Corporation Matthews, NC, USA). A three-stage dilution was conducted in a closed system using 5 mL of 65% HNO_3_. After mineralization, samples were transferred to 10 mL flasks filled with deionized water. An ICP-MS was used to determine the concentration of elements in the mineralized plant tissues.

Plant roots, stems, and leaves were collected after 72 h of treatment for the analysis of metal distribution. Samples were cut into 3 mm long pieces and ablated along the pre-defined line across the cross-sections. Laser performance was optimized according to a detailed scheme [47] using a single variable method.

### 4.4. Superoxide Anion Determination

The superoxide anion content was determined according to Doke [48]. *B. juncea* roots (0.5 g) were placed in the test tubes that were filled with 7 mL of a mixture containing 50 mM phosphate buffer (pH 7.8), 0.05% NBT (nitro blue tetrazolium), and 10 mM of NaN_3_. Next, the test tubes were incubated in the dark for 5 min, and then 2 mL of the solution was taken from the tubes, heated at 85 °C for 10–15 min, cooled on ice for 5 min, and the absorbance was measured at 580 nm against the control.

### 4.5. Hydrogen Peroxide Content

The hydrogen peroxide content was determined using the method described by Patterson et al. [49]. The decrease in absorbance was measured at 508 nm. The reaction mixture contained 50 mM phosphate buffer (pH 8.4) and reagents, 0.6 mM 4-(-2 pyridylazo)resorcinol, and 0.6 mM potassium-titanium oxalate (1:1). The corresponding concentration of H_2_O_2_ was determined against the standard curve of H_2_O_2_.

### 4.6. In Situ Detection of Superoxide Anion and Hydrogen Peroxide

The roots and shoots from plants exposed to metals for 24 h were submerged for 12 h in 100 µM of CaCl_2_ containing 20 µM of dihydroethidium (DHE, pH 4.75; samples for superoxide anion radicals) or 4 µM dichlorodihydrofluorescein diacetate (DCFH-DA) (samples for hydrogen peroxide) in 5 mM dimethyl sulfoxide (DMSO). After rinsing with 100 µM of CaCl_2_ or 50 mM phosphate buffer (pH 7.4), the roots and shoots were observed with a confocal microscope (Zeiss LSM 510, Axiovert 200 M, Jena, Germany) equipped with no. 10 filter set (excitation 450–490 nm, emission 520 nm or more).

### 4.7. Estimation of Protein Oxidation

For carbonyl quantification, the reaction with DNPH was used basically as described by Levine et al. [50]. For each determination, two replicates and their respective blanks were used. Roots and shoots (0.5 g) were incubated with isolation buffer containing 0.1 M Na-phosphate buffer, 0.2% (*v/v*) Triton X—100, 1 mM EDTA, and 1 mM PMSF. After centrifugation at 13,000× *g* for 15 min, supernatants (200 µL) were mixed with 300 µL of 10 mM DNPH in 2 M HCl. The blank was incubated in 2 M HCl. After 1 h incubation at room temperature, proteins were precipitated with 10% (*w*/*v*) trichloroacetic acid (TCA), and the pellets were washed three times with 500 µL of ethanol/ethylacetate (1:1). The pellets were finally dissolved in 6 m guanidine hydrochloride in 20 mM potassium phosphate buffer (pH 2.3), and the absorption was measured at 370 nm. Protein recovery was estimated for each sample by measuring the absorption at 280 nm. The carbonyl content was calculated using the molar absorption coefficient for aliphatic hydrazones, 22,000 m^−1^ cm^−1^.

### 4.8. Determination of Antioxidant Enzyme Activities

The activity of SOD was assayed by measuring its ability to inhibit the photochemical reduction of NBT, adopting the method of Beauchamp and Fridovich [51]. The reaction mixture contained 13 μM riboflavin, 13 mM methionine, 63 mM NBT, and 50 mM potassium phosphate buffer (pH 7.8). Absorbance at 560 nm was then measured. One unit of SOD activity has been defined as the amount of enzyme that causes a 50% decrease in the inhibition of NBT reduction. The activity of CAT was determined by directly measuring the decomposition of H_2_O_2_ at 240 nm for 3 min as described by Aebi [52] in 50 mM phosphate buffer (pH 7.0) containing 5 mM H_2_O_2_ and enzyme extract). CAT activity was determined using the extinction coefficient of 36 mM^−1^ cm^−1^ for H_2_O_2_. The activity of APX was assayed using the method described by Nakano and Asada [53] by monitoring the rate of ascorbate oxidation at 290 nm (extinction coefficient of 2.9 mM^−1^ cm^−1^) for 3 min. The reaction mixture consisted of 25–50 μL supernatant, 50 mM phosphate buffer (pH 7.0), 20 μM H_2_O_2_, 0.2 mM ascorbate, and 0.2 mM EDTA.

### 4.9. Isolation of Total RNA and RT-PCR

Roots and above-ground parts (100 mg) of *B. juncea* plants in the presence of trace metals and under control conditions were collected for total RNA isolation. The RNA was isolated with TRIzol reagent and tested spectrophotometrically for purity at 260 and 280 nm. Then, RNA was reverse-transcribed with oligo (dT) primers using the RevertAid Reverse Transcriptase Kit (Thermo Science, Lithuania, European Union) after DNA was treated with DNase I (Thermo Science).

Primer pair sequences were as follows (forward/reverse, gene accession number): gtgattgcttgcagggtttt/cagaatacggaagcaaatgtca, X54844.1 (TUB1), ggagcaagtttggttccatt/aaggttattcggccagattg, U30841.1 (MnSOD), gaacaatggtgaaggctgtg/gtgaccacctttcccaagat M63003.1 (CuZnSOD). 

As a reference gene, the gene encoding tubulin was used. PCRs were performed with 30 (BJMnSOD) and 34 (BjCuZnSOD) cycles of denaturation, 95 °C for 30 s; annealing primers, 53 °C for 30 s; and elongation, 72 °C for 30 s using a 1:100 diluted cDNA template and REDAllegro*Taq* DNA Polymerase (Novazym, Poznań, Poland).

PCR products were separated by electrophoresis on a 1.3% agarose gel with ethidium bromide in TBE (445 mM Tris-HCL; 445 mM boric acid; 10 mM EDTA; pH 8.0), visualized under UV light and photographed using the Photo Print 215SD V.99 Vilber Lourmat Set. CP Atlas 2.0 were used for densitometric analysis of relative gene expression.

### 4.10. Western Blot

RIPA buffer (150 mM NaCl, 1% Triton X-100, 0.5% Na deoxycholate, 0.1% SDS, 50 mM Tris, pH 8.0) was used to lyse cells for protein extraction. The protein concentrations were determined using the Bradford method, and 20 µg of each extract was loaded onto a 12% SDS–PAGE (sodium dodecyl sulfate–polyacrylamide gel electrophoresis) gel. Separated proteins were transferred to polyvinylidene fluoride membrane (ImmobilonTM-P, Millipore, Burlington, MA, USA) at 350 mA for 1 h using the Mini Trans-BlotCell (Bio-Rad, Hercules, CA, USA). Membranes were blocked with 1% BSA and incubated with an antibody against CuZnSOD at a final dilution of 1:2500. The secondary antibody, goat anti-rabbit IgG conjugated with alkaline phosphatase (Sigma-Aldrich, St Louis, MO, USA), was used at a 1:3000 dilution to visualize protein bands by reaction with 5-bromo-4-chloro-3-indolyl phosphate/nitroblue tetrazolium (BCIP/NBT) (Sigma-Aldrich, St Louis, MO, USA/CALBIOCHEMV.S. and Canada) as a substrate.

### 4.11. Protein Quantification

Total soluble protein contents were determined according to the method of Bradford [54] using the Bio-Rad assay kit with bovine serum albumin as a calibration standard.

### 4.12. Statistical Analyses

Each experiment was performed in three biological and technical replicates. The mean values ± SE are given in the tables and figures. The data were analyzed statistically using IBM SPSS Statistics (Version 22 for Windows). Significant differences among treatments were analyzed by one-way ANOVA, taking *p* < 0.05 as the significance threshold, and the b-Tukey post-hoc test was conducted for pairwise comparisons between treatments.

## 5. Conclusions

This study was conducted to determine the interactive role of Pb, Cu, Cd, and Zn in metal uptake, plant growth, and the antioxidative system of *B. juncea*. Plants accumulated high amounts of trace metals, i.e., more than 40% in the roots, and in the above-ground parts, the values for Cu, Cd, Zn, and Pb were 58%, 55%, 52%, and 38%, respectively. The results suggest that *B. juncea* var. Malopolska is a good hyperaccumulator of trace metals, especially Cu, Cd, and Zn, and can be useful in phytoremediation. The presence of metals resulted in a considerable reduction in *B. juncea* biomass; the highest reduction was observed in plants treated with Cu and Cd. Despite the visible influence of trace metals on plant morphology, the IT coefficient was high and exceeded 90%, indicating the high resistance of *B. juncea* plants. Trace metals lead to the production of ROS, which causes an imbalance in the redox state in the plant cells and increases the level of oxidized proteins. We noticed that under the conditions of oxidative stress, the antioxidant system was activated: SOD, CAT, and APX. We observed that the presence of metals influenced the increase in the activity of antioxidant enzymes, while no significant differences were observed in the levels of CuZnSOD and MnSOD transcripts and proteins. The results obtained indicate that *B. juncea* var. Malopolska has efficient defense mechanisms to cope with different metals.

## Figures and Tables

**Figure 1 ijms-20-04355-f001:**
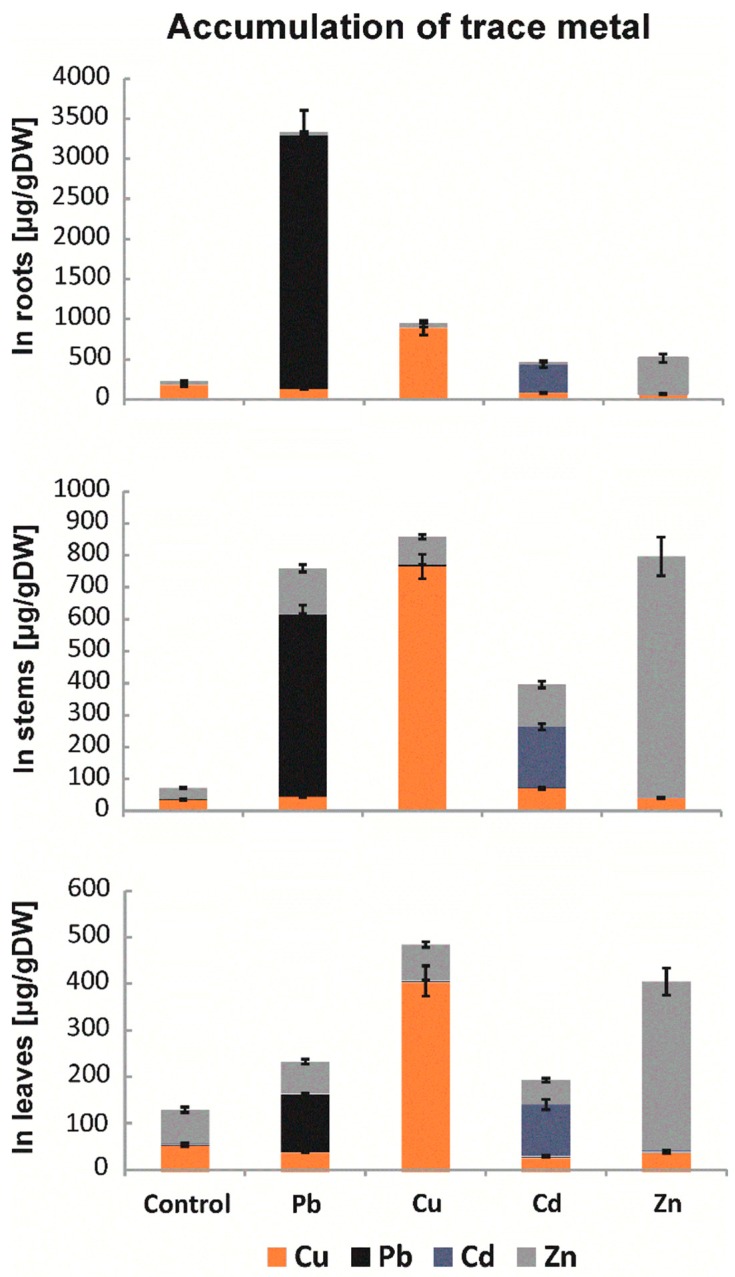
Accumulation of Pb, Cu, Cd, and Zn in the roots, stems, and leaves of *B. juncea* var. Malopolska seedlings grown in Hoagland’s medium and treated with lead, cooper, cadmium, and zinc ions. Metal solutions Pb(NO_3_)_2_, CuSO_4_, CdCl_2_, and ZnSO_4_ were applied at a 50 μM concentration. Mean values of three replicates (±SD).

**Figure 2 ijms-20-04355-f002:**
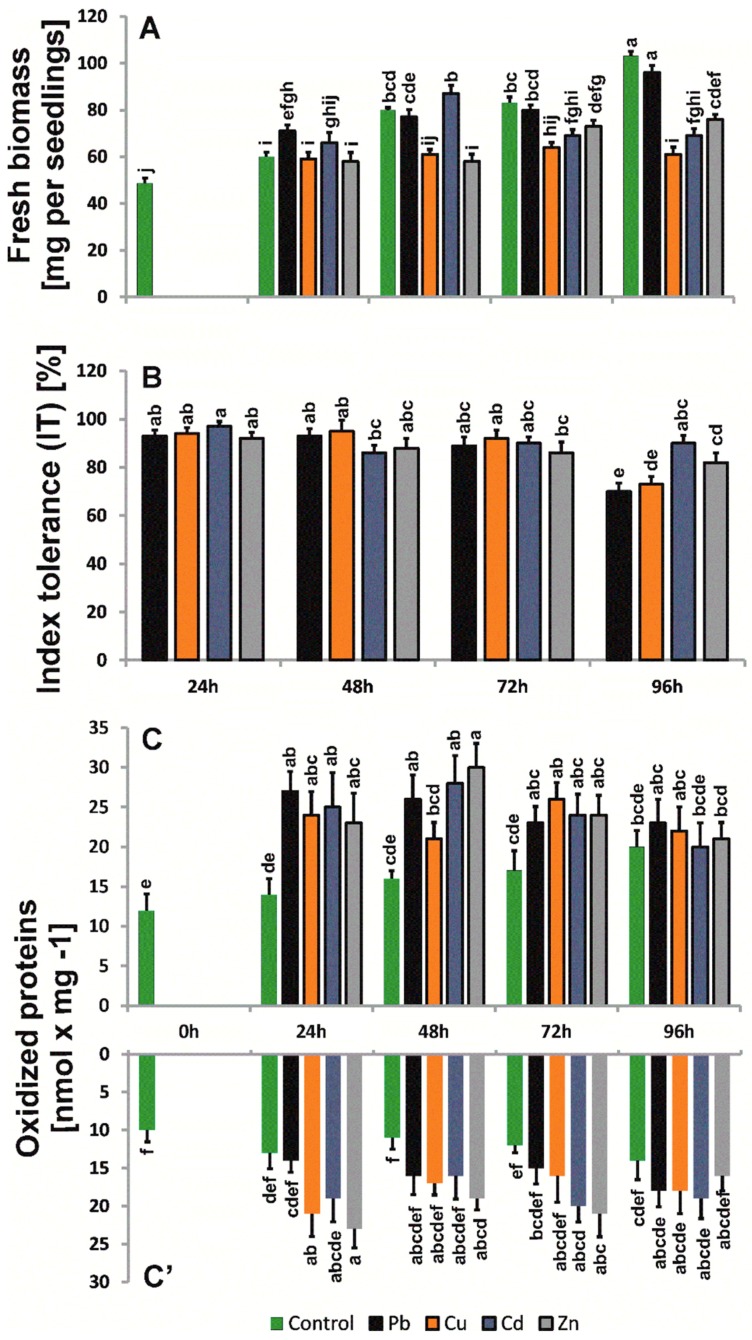
Stress parameters in *B. juncea* seedlings treated with trace metals: Pb, Cu, Cd, and Zn. The results are expressed as the mean ± standard deviation (*n* = 3). Metal solutions Pb(NO_3_)_2_, CuSO_4_, CdCl_2_, and ZnSO_4_ were applied at a 50 μM concentration. Mean values of three replicates (±SD).

**Figure 3 ijms-20-04355-f003:**
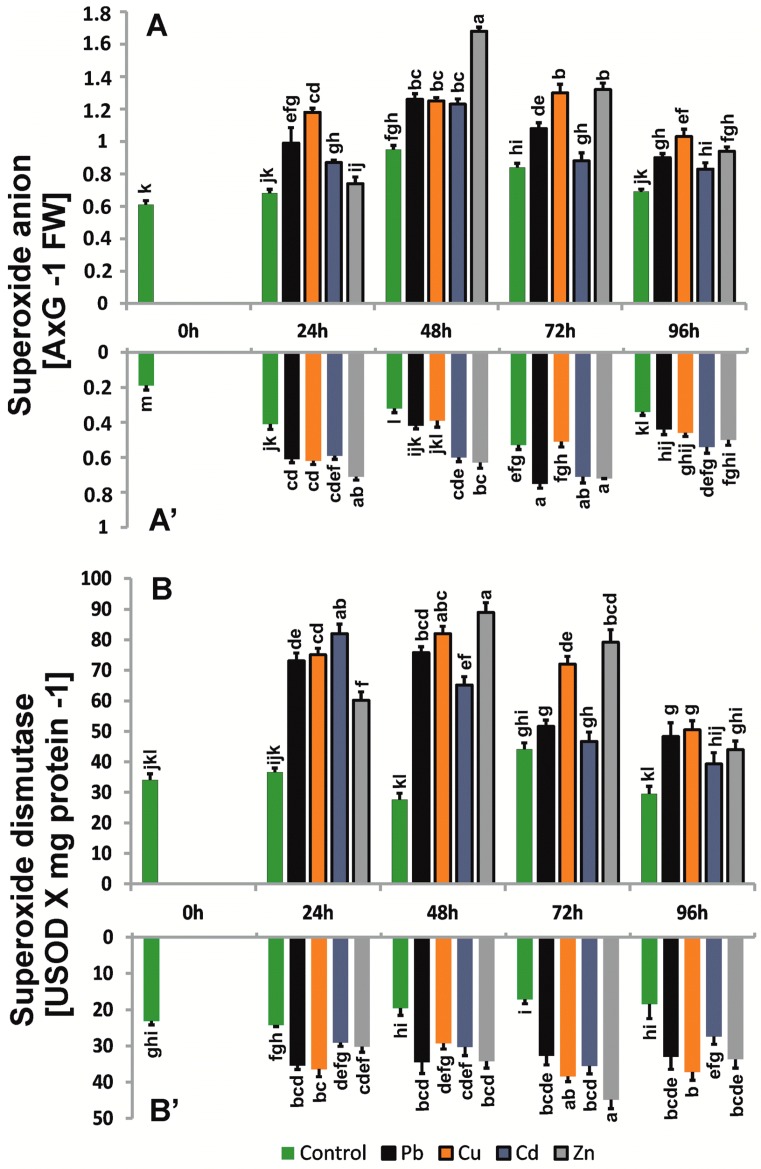
Superoxide anion (A580 g^−1^ FW) level and SOD (USOD mg^-1^ protein) activities in roots and above-ground parts of *B. juncea* var. Malopolska seedlings grown in Hoagland’s medium and treated with lead, cooper, cadmium, and zinc ions. Metal solutions Pb(NO_3_)_2_, CuSO_4_, CdCl_2_, and ZnSO_4_ were applied at a 50 μM concentration. Mean values of three replicates (±SD).

**Figure 4 ijms-20-04355-f004:**
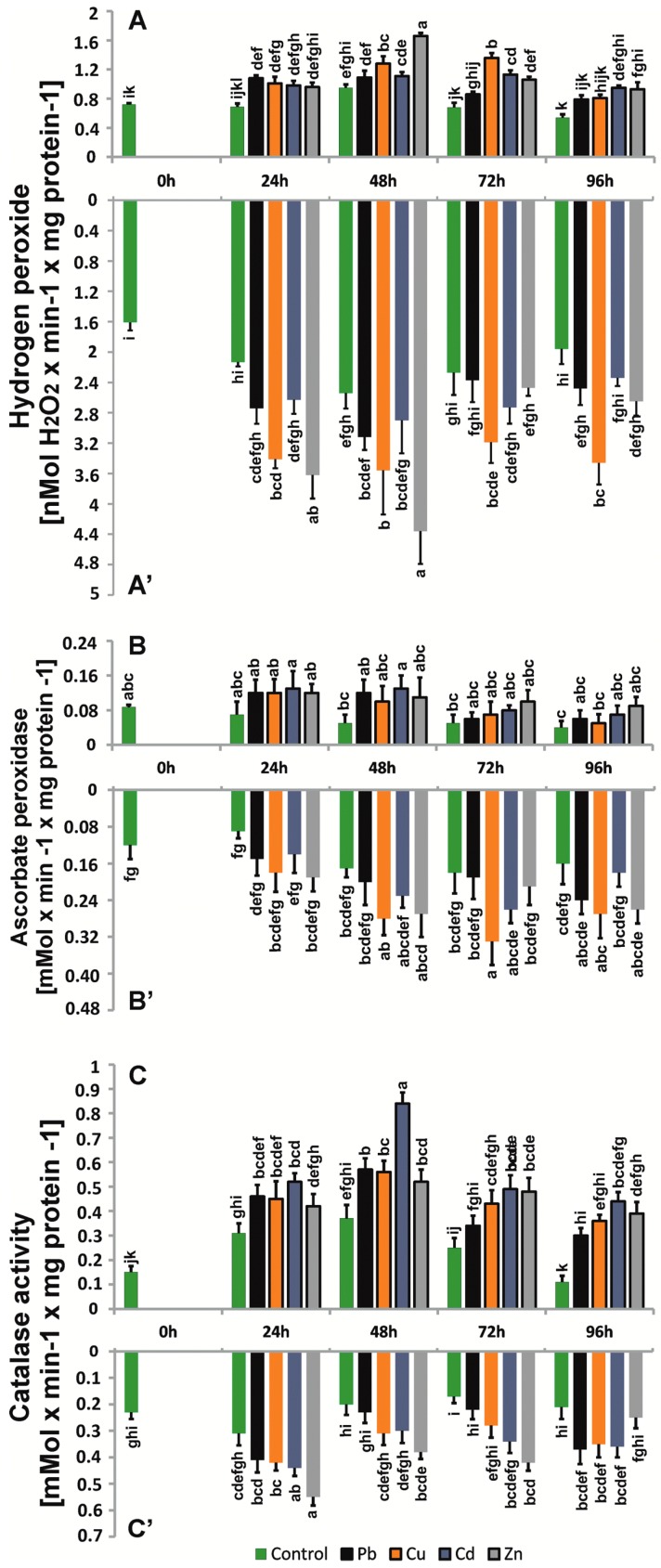
Hydrogen peroxide level (nMol H_2_O_2_ × min^−1^ × mg protein^−1^), CAT (μMol min^−1^ mg^−1^ protein), and APX (μMol × min^−1^ × mg protein^−1^) activities in roots and above-ground parts of *B. juncea* var. Malopolska seedlings grown in Hoagland’s medium and treated with lead, cooper, cadmium, and zinc ions. Metal solutions Pb(NO_3_)_2_, CuSO_4_, CdCl_2_, and ZnSO_4_ were applied at a 50 μM concentration. Mean values of three replicates (±SD).

**Figure 5 ijms-20-04355-f005:**
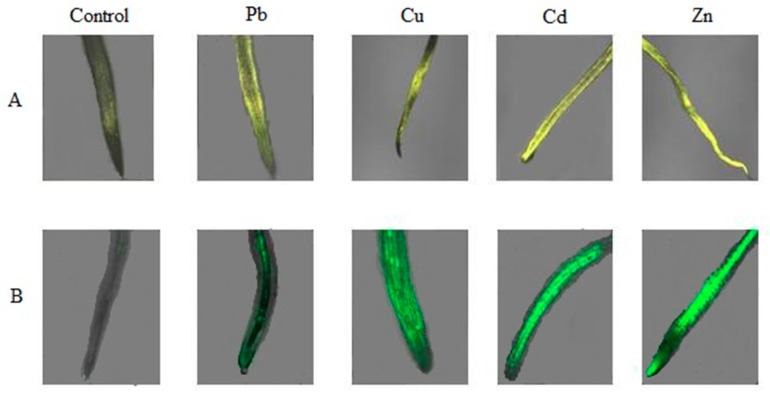
Trace metals-induced O_2_^−^^∙^ and H_2_O_2_ production in *B. juncea* var. Malopolska roots. Fluorescent images of *B. juncea* roots grown in Hoagland’s medium in the presence of 50 μmol of Pb(NO_3_)_2_, CuSO_4_, CdCl_2_, and ZnSO_4_ for 24 h and control roots of plants stained with DHE for 12 h (**A**) and DCFH-DA for 4 h (**B**). The bar indicates 1 μm.

**Figure 6 ijms-20-04355-f006:**
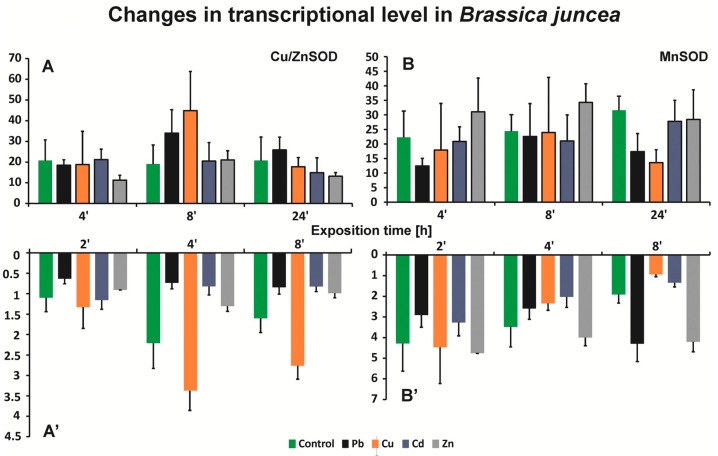
Transcriptional levels of genes encoding antioxidative enzymes in roots and above-ground parts of *B. juncea* var. Malopolska seedlings grown in Hoagland’s medium and treated with lead, cooper, cadmium, and zinc ions. Metal solutions Pb(NO_3_)_2_, CuSO_4_, CdCl_2_, and ZnSO_4_ were applied at a 50 μMol concentration. Enzymes chosen for the experiment were amplified using semi-quantitative RT-PCR with primers designed for *Arabidopsis thaliana* genes: *CSD1* for CuZnSOD and *MSD1* for MnSOD.

**Figure 7 ijms-20-04355-f007:**
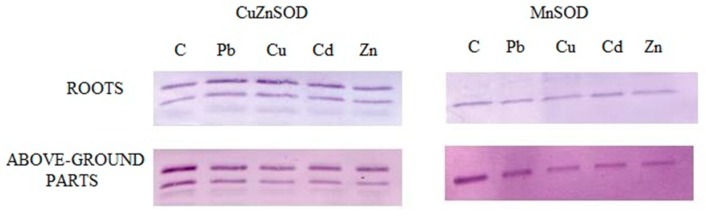
Effects of 50 μM Pb, Cu, Cd, and Zn for 24 h on the CuZnSOD and MnSOD of roots and above-ground parts of *B. juncea* var. Malopolska seedlings. The protein content was evaluated by Western blot using specific antibodies.

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
