# Peer review of "Insight into the Phytoremediation Capability of Brassica juncea (v. Malopolska): Metal Accumulation and Antioxidant Enzyme Activity"

_ijms, 2019, doi:10.3390/ijms20184355_

Round 1
Reviewer 1 Report
The authors have adressed succesfully my concerns, a carefull reading is needed thought to corer some minor linguist mistakes.
Also the authors should conform with the Journals directions regarding the way the references are written in text.
Reviewer 2 Report
I think this paper is now well revised and may be accepted for publication.
Author Response
Response to Reviewer 2,
I think this paper is now well revised and may be accepted for publication.
Answer: Think you for all precious recommendation and suggestions at evaluating our article. Thank you for accepting this manuscript for publication in the Journal.
This manuscript is a resubmission of an earlier submission. The following is a list of the peer review reports and author responses from that submission.
Round 1
Reviewer 1 Report
Reviewer comments:
Manuscript Number ijms-503814
The manuscript entitled: “Insight into phytoremediation capability of Brassica juncea (v. Malopolska): metal accumulation and antioxidant enzymes activity” examines the metal accumulating capability of B. juncea plants together with the development of defense mechanism.
Although the manuscript approaches an interesting subject, it cannot be published in its present form given the following concerns:
1. One basic concern has to do with the level of English used. I would urge the authors to instruct a native English speaker in order to make the proper corrections. The meaning of some sentences is so difficult to follow and that is a petty since the results of the manuscript are quite interesting.
2. What is more is that although the authors state in the material and methods section that a statistical analysis was conducted this is not properly shown on either tables or diagrams. The authors should depict the statistical significance levels on tables or diagrams.
3. Also in the results section the authors state that “We observed in above-ground parts of seedlings the occurrence of necrotic spots on leaves and inhibition of leaf blade surface growth with respect to control seedlings. Moreover, in Cd treated seedlings leaves were slightly twisted whereas Cu caused strong chlorosis and shortening of the tail of leaves. The smallest morphological changes were observed for seedlings treated with Zn”. These descripted morphological changes should appear in text (maybe show some pictures of the leaves and a diagram showing the reduction of the leaf blade surface).
Therefore I would recommend a major revision to be conducted and then the manuscript to be reconsidered
Minor comments:
Table legends usually are above the the table and not beneath it
APOX is usually referred as APX
Acknowledgments usually appear at the end of the manuscript
Reviewer 2 Report
Although the results are interesting this paper does not clearly presents the phytochelation proces. The authors should investigte the associated enzymes and non-enzymating antioxidant (e.g. GSH) to make it complete..
Language quality is poor.
Graphs are poorly organized.